# Addition of Different Levels of Humic Substances Extracted from Worm Compost in Broiler Feeds

**DOI:** 10.3390/ani11113199

**Published:** 2021-11-09

**Authors:** Alejandra Domínguez-Negrete, Sergio Gómez-Rosales, María de Lourdes Angeles, Luis Humberto López-Hernández, Tercia Cesaria Reis de Souza, Juan David Latorre-Cárdenas, Guillermo Téllez-Isaias

**Affiliations:** 1Faculty of Natural Sciences, Autonomous University of Queretaro, Av. de las Ciencias S/N, Juriquilla, Queretaro 76230, Mexico; mvzaledom@gmail.com (A.D.-N.); tercia@uaq.mx (T.C.R.d.S.); 2National Center of Disciplinary Research in Animal Physiology and Genetics, INIFAP, Km 1 Carretera a Colon Ajuchitlán, Queretaro 76280, Mexico; angeles.lourdes@inifap.gob.mx (M.d.L.A.); lopez.lhumberto@inifap.gob.mx (L.H.L.-H.); 3Department of Poultry Science, University of Arkansas, Fayetteville, AR 72701, USA; jl115@uark.edu (J.D.L.-C.); gtellez@uark.edu (G.T.-I.)

**Keywords:** broilers, performance, humic substances, lactic acid bacteria, antioxidant

## Abstract

**Simple Summary:**

In recent years, humic substances (HS) have been tested as growth promoters and health enhancers in poultry with promising results. The main available sources of HS are obtained from leonardite mines; however, HS can also be extracted from organic matter that has undergone a humification process such as compost and worm compost prepared with animal manure. The present study was designed to evaluate the inclusion of different levels of HS extracted from worm compost in the feeds of broilers raised in floor pens from 1 to 42 days of age. The main benefits found in broilers fed HS were the lower feed conversion ratio and lower mortality from 1–42 days, and increased lactic acid bacteria counts in the jejunum of ten-day-old broilers compared to broilers fed diets containing an antibiotic growth promoter and an anticoccidial product. The results indicate that HS extracted from worm compost have a similar effect to HS from leonardite when added to broiler feeds.

**Abstract:**

Different sources and inclusion levels of humic substances (HS) have been tested in broiler rations as an alternative to the addition of growth promoter antibiotics (GPA) with promising results. The current study was carried out to assess the influence of HS extracted from worm compost on broiler production parameters, carcass yield, tibia characteristics, lactic acid bacteria (LAB) counts, excretion of *Eimeria* oocysts, and antioxidant status of breast meat. A total of 1200 broilers were used, housed in groups of 30 per pen, and assigned to five treatments: 1 = basal diet with GPA (positive control), 2 = basal diet without GPA (negative control), 3–5 = basal diet with 0.15, 0.30, and 0.45% HS, respectively. The data was subjected to a variance analysis and orthogonal contrasts. The FI decreased linearly (*p <* 0.05) from 1–14, 29–42, and 1–42 days as the inclusion of HS in the feed increased. The FCR had quadratic responses (*p <* 0.01) from 29–42 and 1–42 days concerning the HS inclusion levels. Lactic acid bacteria was higher (*p ˂* 0.05) in ten-day-old chicks with 0.45% HS in the diet. The 1,1-diphenyl-2-picrylhydrazyl radical scavenging activity antioxidant potential decreased linearly (*p <* 0.05) concerning increasing HS in the feed. The results indicate that HS can be used as growth promoters in broiler feeds.

## 1. Introduction

Humic substances (HS) are complex molecules that originate from the humification process of decaying organic matter, particularly plants, and are natural constituents of water, soils, and lignite [1,2,3]. It has been documented that HS have anti-inflammatory, antibacterial, antiviral, and antitumor properties in humans and animals [1,4,5]. For centuries, HS have been used in humans as nutritional supplements and for therapeutic purposes.

The earliest descriptions of the medicinal applications of HS can be found in Sanskrit and ancient writings from China and Rome, where mystical properties were attributed to them [5,6]. Some of the medicinal properties of HS are described in the Pharmacological Compendium of Chinese Materia Medica, dating back to the 15th century Ming Dynasty; its medicinal use was approved by the Drug Administration in China, where HS are used for the treatment of a wide range of diseases and were called the “golden medicine” [7]. The balneotherapeutic use of peat in ancient Babylon, in Lower Mesopotamia, and the Roman Empire is another of the most significant medical applications of HS [8]. In more recent years, medical aspects, results of successful preclinical and clinical trials, and therapeutic results of HS application in humans have been published [4,5,9].

In the early 1800s, the characterization and chemical description of HS was first carried out by Berzelius [10], one of the predecessors of modern chemistry. HS are mainly composed of humic acids (HA), fulvic acids (FA), and humins; the most concentrated commercial sources of HS are leonardite and lignite, which are obtained from mines, are used extensively to improve crop yields in agriculture; to a lesser extent, they are used as growth promoters in animals [2,3,11]. HS can also be found in concentrations between 8–12% in composts and worm composts prepared from different sources of organic matter, including manure from farm animals [12].

Several benefits on growth and physiology have been reported in broilers supplemented with various sources of HS [13,14,15,16]. The addition of a polymeric polyhydroxy acid mixture (humic, fulvic, ulmic, and humatomelanic acids extracted from leonardite (Farmagulator DRY^TM^ Humate) improved body weight from 21–42 days and feed conversion ratio (FCR) from 21–42 and 1–42 days of age. [13]. In another study using the same source of HS [(Farmagulator DRY^TM^ Humate] increased body weight and intestinal villus length, as well as lower FCR and intestinal crypt depth were found [14]. Improved body and carcass weight, as well as lower FCR and blood cholesterol, were observed in broilers fed HS extracted from leonardite containing polymeric polyhydroxy acids with a ratio of 2.08 HA to FA as active forms [15]. Furthermore, the addition of potassium humates to broilers increased the Ca and P content of the tibia, the length and width of the intestinal villus, and the distribution and density of lymphoid tissue in the bursa of Fabricius and thyme [16].

In addition, in broilers added with liquid HS and an extract of HS from worm compost, higher productivity and retention of energy, nitrogen, and ashes, as well as increased carcass yield and lactic acid bacteria (LAB) counts in the intestine and reduced coccidial oocyst excretion were observed [17,18]. These findings suggest that HS extracted from worm compost obtained from animal manure and added in broiler water or feed can improve productivity and health as has been observed with the use of commercial sources of HS. In a previous study, broilers kept in metabolic cages were supplemented with HS extracted from worm compost from 14 to 35 days of age [18]; however, the addition of HS from worm compost has not been evaluated in broilers raised in floor pens throughout the entire productive cycle. The extent to which different doses of HS from worm compost can improve different response variables, as observed for body weight and FCR in chickens supplemented with HS from leonardite [14], is also unknown. Therefore, the objective of the study was to evaluate the productive variables, carcass yield, tibia characteristics, LAB count, *Eimeria* oocyst excretion, and meat antioxidant status in broilers supplemented with increasing levels of HS from worm compost in the feeds.

## 2. Materials and Methods

### 2.1. Characterization of the HS

HS were extracted from a worm compost prepared with sheep manure using the al kaline extraction methodology previously recommended [19]. The sheep manure went through a two-week precomposting process before being inoculated with Red Californian worms and harvested three months later. The worm composting was done in a greenhouse with natural ventilation.

Before the extraction, foreign materials were removed from the worm compost, dried at room temperature, and passed through a metal mesh to homogenize the particle size. HS extraction was carried out in three phases. In the first phase, 100 kg of dry worm compost were deposited in a trough and were mixed with 400 L of sodium hydroxide (NaOH, 0.5 M), the mixture was homogenized and allowed to stand for 24 h. The mixture was decanted to separate the liquid fraction, which was collected in a container. In the second phase, 300 L of 0.1 M NaOH were added to the remaining solid fraction, homogenized and after 24 h the liquid fraction was decanted and collected in the same container. In the third phase, 100 L of water were added to the remaining solid fraction, homogenized, and allowed to stand for 24 h again. The mixture was decanted and the liquid fraction was stored in the container. The three extracted liquid fractions were mixed and homogenized, deposited in drying troughs covered with plastic, inside a greenhouse with natural ventilation. Fifty liters of extract were deposited in each trough and left to dry at room temperature for seven days. At the end of this period, the solid was collected and dried for 48 h at 55 °C in a forced-air oven (Shel Lab, Cornelius, OR, USA) before being ground in a Thomas Willey mill with a 0.1 mm diameter screen.

A sample of the liquid extract was taken to determine the concentration of HA and FA following the procedure described previously [19]. In 100 mL beakers, 50 mL of the liquid extract and 50 mL of 10% sulfuric acid were added and allowed to stand for 24 h. The liquid fraction containing FA was separated from the precipitated fraction containing HA by decantation; the two fractions were subjected to three washes in a water bath with distilled water, and at the end, both fractions were dried in an oven at 100 °C (Terlab S.A de C.V., Zapopan, Jalisco, Mexico) and the ash concentration was calculated after 6 h in a furnace (Furnatrol I Type 1,8200; Thermolyne, Guadalajara, Jalisco, Mexico) at 600 °C. The difference between the weight of dry matter and ashes was used to estimate the concentration of HA and FA. The concentration of functional groups, elemental analysis, types of crystals, and the percentage of aromaticity in HS were previously published [18].

### 2.2. Animals, Treatments, and Diets

The current study was approved with folio number 108FCN2018 by the Bioethics Committee of the Faculty of Natural Sciences of the Autonomous University of Queretaro. Twelve hundred one-day-old male Ross 308 chicks were used. The environment of the facilities was manually controlled, with canvas curtains and gas brooders. During rearing, the temperature was initially set at 32 °C and gradually decreased at a rate of 2 °C each week until reaching 26 °C after 21 days. The lighting program was 23 h of light for 1 h of darkness for the first seven days, and from day eight on, 20 h of light for 4 h of darkness were used. Water and feed were freely available during the whole production trial.

At arrival, broilers were housed in 40-floor pens of 1 × 1.5 m equipped with a bell drinker and a hanging hopper feeder; a layer of five cm of sawdust was placed on the floor of each pen. Thirty broilers were housed in each pen and were randomly assigned to five treatments: (1) A positive control basal diet, added with a growth promoter antibiotic (GPA) and coccidiostat, (2) negative control diet without GPA or coccidiostat, and (3–5) negative control diet supplemented with 0.15, 0.30, and 0.45% HS, respectively. Diets were fed in mash form the whole production trial and the ingredient and nutrient composition are shown in Table 1. In the positive control diet, bacitracin methylene disalicylate (BMD) was used as GPA at a dose of 55 g/ton of feed and nicarbazin was used as coccidiostat from 1 to 21 d at a dose of 125 g/ton of feed, and salinomycin from 22 to 42 d at a dose of 60 g/ton of feed. HS were added at the top of the diets. The concentration of HA, FA, and ash in the dry HS was 43.5, 29.0, and 27.5%, respectively, and the estimated aromaticity was 53.8% [18]. The productive cycle was divided into three periods: starter from 1–14, growing from 15–28, and finishing from 29–42 d. On day 1 and day 42 of the study, the stocking density was 15 and 10.5 broilers/m^2^, respectively.

### 2.3. Sample Collection and Laboratory Determinations

Chicks were weighed on days 1, 14, 28, and 42 of age to estimate the daily weight gain (WG, g/d). The amount of feed offered and rejected per period was recorded in order to estimate the feed intake per day (FI, g/d). By dividing FI/WG, the FCR was estimated. When the study was concluded, the overall mortality was also registered and expressed in percentage. On days 10, 24 and 38 of age, three chicks from each pen were slaughtered by cervical dislocation. The breast and carcass were weighed and their yield was estimated by dividing their weight between the body weight and the result was expressed as a percentage. The left tibia was obtained from each chicken and kept in individual plastic bags and stored at −20 °C. Samples of jejunum contents were taken in sterile bags for LAB counts. Jejunal contents were immediately transported to the lab for culturing. Also, on days 10, 24, and 38 of age, two broilers per pen were housed during 1 h in individual cages provided with trays and fecal samples were collected for *Eimeria* oocyst counts. Feces were collected with sterile scoops and placed in wirl-pack bags with a 50/50 ratio of potassium dichromate solution. At the end of the experiment, at 42 days of age, three broilers per pen were slaughtered and the breast was removed to determine the antioxidant capacity of the meat.

The dry matter in the tibias were determined after 24 h in a horizontal flow hot-air oven (Terlab S.A. de C.V., Zapopan, Jalisco, Mexico) at 105 °C, and the ashes were measured in a furnace (Furnatrol I Type 1,8200; Thermolyne, Guadalajara, Jalisco, Mexico) after 6 h at 600 °C. Glass tubes prepared with 0.01% peptonized water and serial dilutions of 1:1–7 wt/vol were used to determine LAB counts in the jejunum contents. From each tube 100 µL were taken and placed in Petri dishes prepared with Man, Rogosa and Sharpe agar (MRS DIBCO S.A de C.V). They were kept at 35 °C in an anaerobic environment for 48 h using anaerobiosis containers and microaerophilic envelopes to kept the atmosphere at 5% O_2_ (GasPak Ez, BD Diagnostics, Sparks, MD, USA). Log_10_ colony forming units (CFUlog_10_/g) were used to express the results.

The *Eimeria* oocyst count was done using 2 g of feces deposited in 50 mL centrifuge bottles with 40 mL of supersaturated saline solution. The bottles were slightly shaken and an aliquot was taken to fill the McMaster chamber (CHALEX Corp., Salt Lake City, MD, USA). Oocysts per cell were quantified with objective lenses of 10 magnifications and the result was multiplied by 100 and divided by two to be expressed as oocysts/g feces.

The breasts were processed on the day of slaughter. They were first placed in a cold room until they reached 4 °C. A HI 99,163 meat pH meter connected to a glass puncture electrode was used to measure the pH (HANNA Instruments Mexico, Mexico City, Mexico). Two methodologies were used to evaluate the water holding capacity. The drip loss [20] was assessed by placing 100 g breast in a sealed polyethylene bag. The samples were reweighed after being stored at 4 °C for 24 h. The difference in the weight of the sample before and after chilling is reported as a percentage of drip loss. For the centrifugation method [21], a tube containing 5× *g* of minced meat was placed and 8 mL of NaCl was added while the tube was agitated for 1 min. After 30 min in a cold water bath, the tubes were centrifuged (Eppendorf 5810R centrifuge, Hamburg, Germany) at 12,000× for 15 min. In a 10 mL tube, the supernatant was measured and by difference, the percentage of NaCl retained in the flesh was quantified.

The antioxidant activity was measured using two techniques. The scavenging activity of 1,1-diphenyl-2-picrylhydrazyl radicals (DPPH) [20] and the antioxidant power of ferric radicals (FRAP) [22] were quantified in 5 g of meat extracts. The sample was first homogenized for 1 min in 25 mL of phosphate buffer (IKA T25 homogenizer), then centrifuged for 30 min at 12,000× *g* at 4 °C before being filtered on Whatman filter (No. 4) paper. The filtered extract was stored at −20 °C in Eppendorf tubes. The DPPH determinations were performed, in 25 µL aliquot of the extract mixed in test tubes with 975 µL of the DPPH solution. After 1 h incubation at room temperature in a dark room, the absorbance was read at 515 nm using a UV/VIS spectrophotometer (GENESYS 10S UV-Vis Thermo Scientific). The trolox equivalents in mmol/kg meat were used to express the results. The FRAP was analyzed in 25 µL aliquot of the extract blended in test tubes with 975 µL of a solution of FRAP containing 2.5 mL 2,4,6-tripyridyl-s-triazine acid (TPTZ) 40 mM, 2.5 mL FeCl3 20 mM, and 25 mL acetate buffer 0.3 mM, pH 3.6. As before, the absorbance was measured at 593 nm, and readings were taken at 0 and 6 min after the reaction started. Trolox equivalents in mmol/kg of meat were calculated to show the results.

The thiobarbituric acid reactive substances (TBARS) test [23] was used to evaluate the lipid oxidation. In the first step, an ice bath with 20 mL of trichloroacetic acid (TCA) was used to homogenize 5 g of meat for 1 min. During 20 min at 12,000× *g*, the homogenate was centrifuged at 4 °C. The floating extract was filtered and stored at −20 °C in tubes protected from light. In test tubes, 1 mL of TBA and 1 mL of the extract were mixed to quantify the TBARS. During 30 min the tubes were heated at 95 °C, and the absorbance was read at 530 nm after centrifugation. Malondialdehyde (MDA) in mg/kg meat were estimated to explain the results.

### 2.4. Statistical Analysis of Data

Analysis of variance was performed using a complete randomized model [24] for the collected data. A statistical analysis was performed for each response variable in each growth phase and on each day of slaughter, using five treatments in the model (positive control; negative control; 0.15, 0.30, and 0.45% HS). The method of the least significant difference was used to determine the difference between treatments. Values in percentage were transformed to arcsin before analysis. The type of response concerning the dietary addition of HS (linear, quadratic, or cubic) was obtained through polynomials contrasts.

## 3. Results

The productive variables of broilers fed increasing levels of HS are shown in Table 2. The body weight at day 14, 28, and 42, as well as the WG, FI and FCR from 1–14 and 15–28 days, as well as the WG from 29–42 and 1–42 days were similar among treatments. From 29–42 days, broilers in the positive control and negative control had similar FI, and the FI of these groups was higher (*p* ˂ 0.05) than that of broilers receiving 0.15, 0.30, and 0.45% HS. The FCR was lower (*p* ˂ 0.01) in the positive control compared to the negative control birds, but was higher to the group with 0.15% HS, and similar to the groups with 0.30 and 0.45% HS. From 1–42 days, the FI was higher in the positive and negative controls (*p* ˂ 0.01), was intermediate at 0.15% HS and was lower at 0.30 and 0.45% HS. The FCR was lower (*p* ˂ 0.01) in the positive control compared to the negative control, but was higher compared to the 0.15% HS and was similar to the 0.30 and 0.45% HS groups. The overall mortality was highest (*p* ˂ 0.05) in the negative control, intermediate in the positive control, and lowest at 0.15, 0.30, and 0.45% HS.

At 10, 24, and 38 days of age, the breast and carcass weights and yields were similar among treatments (Table 3). Similarly, there were no differences in tibia dry matter and ashes content at 10, 24, and 38 d (Table 4). The LAB counts (Table 5) at day ten were similar among the positive control, the negative control, and the groups receiving 0.15 and 0.30% HS, but the counts were higher with the addition of 0.45% HS. There were no differences in LAB at 24 and 38 days. There were also no differences among treatments on the oocyst counts at 10, 24, and 38 days (Table 5). No differences among treatments were found on the pH, water holding capacity, FRAP, and TBARS (Table 5). The DPPH antioxidant activity was higher (*p* < 0.05) in the negative control group, intermediate in the positive control and with 0.30% HS, but was lower with 0.15 and 0.45% HS.

The results of polynomial contrasts indicate that from 1–14 days, the FI decreased linearly (*p* ˂ 0.05) as the dietary HS increased. From 29–42 days, the FI and FCR had linear falls (*p* ˂ 0.05) and quadratic (*p* < 0.01) responses due to the increasing dietary levels of HS. From 1–42 days, the FI showed a linear drop (*p* ˂ 0.05) and the FCR had linear (*p* ˂ 0.05), quadratic (*p* ˂ 0.01) and cubic (*p* < 0.05) responses regarding the ascending concentrations of dietary HS. The mortality showed a decreasing linear (*p* ˂ 0.01) and quadratic (*p* ˂ 0.05) responses, the LAB counts exhibited a linear ascending response (*p* < 0.05) and the DPPH radical had a descending linear response (*p* < 0.05) in respect to the raising addition of HS in the feeds.

## 4. Discussion

Broilers added with increasing amounts of HS in the feed showed reduced FI at 1–14, 29–42, and 1–42 days; however, the body weight and WG were similar among treatments, leading to reductions in FCR at 29–42 and 1–42 days of age. In two previous studies, reduction in FI in broilers fed increasing amounts of HS in the drinking water [25] or in the feed were observed [14], which are consistent with the findings of the present study. In other reports, no changes in FI due to the addition of different levels of HS in the water or feed [13,14,17] were reported. But in all these reports, decreased FCR in broilers supplemented with HS was found [13,14,15,17,25]. In two of these reports, there were significant reductions of the FCR in the second part of the trials (22–42 d) and throughout the whole experiments [13,15], which closely resembles the findings of the present research.

The lack of response in productive variables to the addition of HS from worm compost HS from 1–14 and 15–28 days is unknown. Using a liquid source of HS from worm compost in 21–42-day-old broilers, improvements in FCR and nutrient retention were found in metabolic cages and improvements in WG and FCR were seen in floor pens [17]. In a subsequent study, adding pasteurized liquid HS from worm compost to drinking water, higher breast yield in broilers from 21–45 days reared in floor-pen was found [26]. These findings show that using a liquid source of HS from worm compost improves productivity in 21–45-day-old broilers raised in cages or on in floor-pens. This corresponds to the greater benefit on the FCR observed in the current study from 29–42 days, and which caused a carry-over effect on the FCR from 1–42 days of age. In the two previous studies where HS from leonardite was used, there was no response in the productive variables in chicks from 1–21 days; but improvements in body weight and FCR from 22–42 days, and a carry-over effect in these variables from 1–42 days of age were also reported [13,15]. In the current study, it is unknown whether improved FCR from 29–42 days was dependent on, or independent of, the addition of HS from 1–14 and 15–28 days. This topic deserves further clarification in future research.

In some reports, increases in body weight and WG have been observed in broilers supplemented with HS [13,14,15], which was not the case in the present investigation. In some of the mentioned reports, higher carcass or breast yield was reported [15,25]. Higher carcass yield in broilers added with HS extracted from a worm compost was also observed [18]. In the present study, due to the reduction in FI with increasing dietary HS, highly significant linear reductions in ME and digestible lysine intake were also found from 1–14, 29–42, and 1–42 days (data not shown), which could explain the lack of effect of HS on body weight, WG, breast and carcass weight and yield.

In terms of overall mortality, the negative control group, which did not include GPA or HS, had a higher number of dead birds, whereas the positive control and HS treatments had similar mortality. Three previous studies [2,13,26] found a trend of lower mortality in HS-fed chicks compared to positive control groups, though the differences were not statistically significant in any of the cases. These findings are consistent with those of the current study. HS has been shown to stimulate the immune response in broilers [1,2,3]. For example, elevated lymphoid tissue distribution and density in the bursa of Fabricius and thyme [16], increased concentration of antibodies against IBD [26], and greater lymphocytes and leukocytes counts, globulins (α, β, and γ), phagocytosis, and phagocytic index have been found in HS-fed broilers [27]. The trend of lower mortality in HS-added broilers is probably due to a better immune response.

HS are considered the natural ligands with the highest complexation capacity, giving them a strong potential to form chelates with various ions, which has been linked to better mineral utilization in plants and animals [11]. The lack of effect of HS on the dry matter and ashes content of tibia does not agree with previous results in broilers supplemented with HS that had increased tibia ashes content [16,28,29] and Ca, Fe, and Cu concentration in the meat [15,25]. A higher percentage, thickness, and hardness of the eggshell have also been reported in laying hens and pheasants supplemented with HS [30,31,32]. In addition, the supplementation of a worm compost leachate in the drinking water of broilers as a source of HS increased the ashes retention from 18 to 39% [17]. It is probably that the lack of effect of HS on the tibia response variables in the present work was due to the highly significant, linear reductions in calcium, and available phosphorus intake (data not shown), and other minerals due to the reductions of FI in HS-fed broilers.

Information from soil and plant sciences indicate that HS have beneficial effects on various microorganisms [33] including bacteria of the phylum Firmicutes, such as LAB [34] commonly found in the digestive tract of animals. The most outstanding mechanisms of HS on the activity of microorganisms are (a) as a source of substrates, providing carbon, nitrogen, phosphorus, trace elements, and vitamins [35], and (b) as natural surfactants, increasing the permeability of cell membranes in bacteria, due to their amphiphilic character, which enhances the absorption of nitrogen and other micronutrients [36,37]. LAB predominates in most of the digestive tract of broilers, and are normally used as probiotics in the prevention or treatment of intestinal disorders [38,39] because they can control or reduce the growth of potentially pathogenic bacteria through competitive exclusion mechanisms. The increase in LAB counts in broilers killed at ten days agrees with previous results in broilers fed HS from leonardite [40] and from a worm compost [18]. However, the lack of effect of HS on the LAB counts in broilers at 24 and 38 days is unknown. In other studies, no changes in LAB counts were reported in the cecum contents of broilers supplemented with HS from leonardite [41] and in the small intestine of broilers supplemented with HS obtained from worm compost [42].

The lack of effect of HS on oocyst excretion does not coincide with previous reports indicating that HS are effective in reducing parasitic diseases in plants and animals. Examples include the reduction of parasitic infestations in tobacco, lettuce, tomato, bell pepper, grape, and strawberry plants [43,44,45]. In aquaculture, reduced parasitic infections have been reported with the use of HS in goldfish, ornamental fish, common carp, and Nile tilapia [46,47]. In mice experimentally exposed for 21 days to *Trypanosoma brucei brucei* and *T. brucei gambiense*, the addition in the drinking water an extract of humus provided effective protection and reduced the occurrence of deaths [48]. In broilers supplemented with HS, the excretion of coccidial oocysts in feces was reduced compared to broilers that did not receive HS or coccidiostat [18]. In the present study, low numbers of oocysts excreted were observed in all treatments, which indicates a low incidence of coccidia in the experimental farm. This could have nullified the possible reduction oocyst excretion in positive control and HS-added broilers, and also, the increased oocyst excretion in the negative control broilers.

The antioxidant activity of breast muscle as measured by the DPPH radical was reduced at levels of 0.15 and 0.45% HS. The redox properties of HS have been investigated in *in vitro* and *in vivo* studies, with quinones being classified as reducible fractions and phenols being classified as electron-donating fractions with antioxidant properties in comparison to electron-accepting quinones, respectively [49,50]. In *in vivo* studies in broilers supplemented with HS, high increments in glutathione reductase and total antioxidant activity and catalase in blood have been observed [51]. The dietary addition of FA also increased the activity of superoxide dismutase and glutathione peroxidase and reduced the blood levels of malondialdehyde in broilers [52]. Unlike the two aforementioned studies in which the antioxidant activity was assessed in the blood of broilers, in the present research, the antioxidant activity was measured in the breast muscle, as this represents the final product consumed by human, and the shelf life and organoleptic characteristics of meat can be influenced by the antioxidant status of meat [53]. It has also been suggested that the *in vivo* antioxidant effects of HS depend on the composition of their structure; for example, a high concentration of quinones can cause the formation of high amounts of oxygen reactive substances that promote oxidative stress and lipid peroxidation which can overcome the antioxidant response of the muscle. However, it is oxygen-containing functional groups, primarily carboxylic and phenolic groups, that are responsible for enhancing the antioxidant properties of the compound in question [54,55].

It is important to take into account that direct comparison between published studies in broilers supplemented with HS is difficult to make due to different experimental conditions among studies. Some disagreements among studies include the addition of different levels of HS, different concentrations of HA and FA in the tested products, different modes of inclusion such as dry products in feed and liquids in water, and particularly, dissimilarities in the conformational arrangement of the different structures that compose the skeleton and different type, concentration and position of functional groups [17,18,25]. These latter characteristics depend in turn on factors such as the kind of organic matter that gave rise to HS such as decomposing plants and animals, soils, peat, and for the present study, a worm compost, and also on the climatic, geographic, and physical variables, among others, of the site of origin [11,56]. The age or duration of the humification process is another factor that influences the composition of HS since it has been reported that some lignites originated in the Early and Middle Miocene period (approximately 20 million years ago) and are considered mature HS [57]; whereas HS extracted from a worm compost are still in an early humification process and are considered immature [58]. Additionally, conformational changes in the structure or composition of HS can be caused due to the extraction methodology [59].

## 5. Conclusions

The results of the present study indicate that broilers supplemented with increasing levels of HS had reduced FI in different production periods, but similar WG compared to the positive control group, which caused the lower FCR from 29–42 and 1–42 d of age. HS-fed broilers showed lower mortality compared to the negative control group. Another beneficial effect of HS was an increase in LAB counts in ten-day-old chicks at a 0.45% inclusion level. A negative effect of HS addition was the lower DPPH antioxidant activity in the breast muscle. The results indicate that HS extracted from worm compost can be used as growth promoter in broiler feeds.

## Figures and Tables

**Table 1 animals-11-03199-t001:** Composition of the experimental diets.

Item	Starter	Grower	Finisher
Ground corn	63.81	65.31	67.41
Soybean meal	29.40	27.00	24.70
Vegetable oil	2.10	3.30	3.80
Calcium ortophosphate	1.70	1.65	1.52
Calcium carbonate	1.60	1.50	1.43
Salt	0.32	0.30	0.28
DL-Methionine	0.24	0.21	0.18
L-Lysine·HCl	0.26	0.21	0.19
L-Threonine	0.08	0.05	0.04
Sodium bicarbonate	0.20	0.20	0.20
Vitamins and minerals ^1^	0.10	0.10	0.10
Choline chloride	0.09	0.07	0.05
Calculated nutrient content			
ME, kcal/kg	3000	3100	3200
Digestible Lys, %	1.10	1.00	0.90
Digestible Met, %	0.52	0.47	0.43
Digestible Thr, %	0.71	0.65	0.60
Calcium, %	1.00	0.90	0.80
Available phosphorus, %	0.50	0.45	0.40

^1^ Each kg provided: 6500 IU Vit A; 2000 IU Vit D3; 15 IU Vit E; 1.5 mg Vit K; 1.5 mg thiamine; 5 mg riboflavin; 35 mg niacin; 3.5 mg pyridoxine; 10 mg pantothenic acid; 1500 mg choline; 0.6 mg folic acid; 0.15 mg biotin; 0.15 mg Vit B12; 100.0 mg Mn; 100 mg Zn; 50 mg Fe; 10 mg Cu; 1.0 mg I.

**Table 2 animals-11-03199-t002:** Productive performance of broilers added with increasing levels of humic substances in the feed, probability level, and type of effect associated with humic substances.

		Level of Humic Substances, %						Effect of Humic Substances		
Response Variables ^a^	Positive Control	0.00	0.15	0.30	0.45	SEM ^b^	*p*-Value	Lin	Quad	Cub
Body weight, g										
Day 1	44.5	44.7	44.5	44.3	44.0	0.259	0.33	0.24	0.94	0.92
Day 14	418.4	432.9	431.6	426.3	421.6	6.122	0.40	0.19	0.80	0.88
Day 28	1323	1327	1313	1295	1314	15.504	0.65	0.44	0.31	0.55
Day 42	2600	2497	2555	2491	2510	25.682	0.29	0.74	0.28	0.14
Productive performance from 1–14 days of age										
Feed intake, g/d	40.5	41.9	41.6	40.8	40.1	0.573	0.17	0.05	0.78	0.78
Weigh gain, g/d	26.7	27.7	27.6	27.3	27.0	0.436	0.42	0.22	0.80	0.88
Feed conversion ratio	1.52	1.51	1.50	1.50	1.49	0.024	0.93	0.48	0.94	0.94
Productive performance from 15–28 days of age										
Feed intake, g/d	122.1	118.4	119.0	118.8	119.1	1.658	0.53	0.80	0.95	0.86
Weigh gain, g/d	64.6	63.8	63.0	62.1	63.8	1.062	0.51	0.81	0.23	0.57
Feed conversion ratio	1.89	1.86	1.89	1.92	1.87	0.026	0.58	0.65	0.17	0.60
Productive performance from 29–42 days of age										
Feed intake, g/d	218.5 ^c^	215.8 ^c^	204.5 ^d^	199.3 ^d^	203.8 ^d^	3.097	0.05	0.05	0.01	0.44
Weigh gain, g/d	91.2	83.6	88.7	85.4	85.4	1.582	0.12	0.47	0.10	0.23
Feed conversion ratio	2.40 ^c^	2.58 ^d^	2.31 ^e^	2.34 ^c,d^	2.39 ^c,d^	0.045	0.01	0.05	0.01	0.10
Productive performance from 1–42 days of age										
Feed intake, g/d	127.0 ^c^	125.4 ^c,d^	121.7 ^d,e^	119.6 ^e^	121.0 ^e^	1.458	0.01	0.05	0.10	0.79
Weigh gain, g/d	60.8	58.4	59.8	58.3	58.7	0.577	0.29	0.72	0.28	0.14
Feed conversion ratio	2.09 ^c,d^	2.15 ^c^	2.04 ^d^	2.05 ^d^	2.06 ^d^	0.019	0.01	0.05	0.01	0.05
Overall mortality, %	7.08 ^c,d^	13.33 ^c^	5.42 ^d^	1.67 ^d^	3.75 ^d^	2.370	0.05	0.05	0.01	0.86

^a^ Data are means of eight replications (pens) per treatment. ^b^ Standard error of the mean. ^c–e^ Values within rows with different superscripts differ significantly, *p ˂* 0.05.

**Table 3 animals-11-03199-t003:** Breast and carcass weight and yield of broilers added with increasing levels of humic substances in the feed, probability level, and type of effect associated with humic substances.

		Level of Humic Substances, %						Effect of Humic Substances		
Response Variables ^a,b^	Positive Control	0.00	0.15	0.30	0.45	SEM ^c^	*p*-Value	Lin	Quad	Cub
10 days of age										
Breast weight, g	48.1	50.7	48.7	48.4	48.8	1.721	0.84	0.41	0.91	0.45
Breast yield, %	17.6	17.2	17.1	16.9	17.3	0.271	0.37	0.94	0.24	0.53
Carcass weight, g	118.2	127.6	122.4	122.0	122.1	3.686	0.52	0.30	0.46	0.79
Carcass yield, %	43.6	43.3	42.9	42.6	43.2	0.464	0.61	0.77	0.27	0.67
24 days of age										
Breast weight, g	239.4	231.0	237.5	234.0	225.8	6.301	0.57	0.50	0.24	0.85
Breast yield, %	22.1	21.1	21.4	21.6	21.3	0.303	0.23	0.57	0.42	0.79
Carcass weight, g	572.8	566.5	582.6	550.6	545.5	14.765	0.36	0.16	0.48	0.27
Carcass yield, %	52.8	51.8	52.4	51.0	51.5	0.806	0.55	0.60	0.95	0.34
38 days of age										
Breast weight, g	566.7	548.9	553.4	540.4	563.3	14.283	0.50	0.38	0.35	0.34
Breast yield, %	23.3	22.6	22.6	22.2	22.6	0.358	0.23	0.48	0.21	0.35
Carcass weight, g	957.8	947.9	949.7	927.8	977.0	19.670	0.26	0.25	0.13	0.22
Carcass yield, %	39.4	38.9	38.8	38.2	39.3	0.460	0.10	0.15	0.13	0.17

^a^ Data are means of eight replications per treatment with pooled data of three birds in each replication. ^b^ There were no statistical differences among treatments, *p* > 0.05. ^c^ Standard error of the mean.

**Table 4 animals-11-03199-t004:** Tibia dry matter and ashes content of broilers added with increasing levels of humic substances in the feed, probability level, and type of effect associated with humic substances.

		Level of Humic Substances, %						Effect of Humic Substances		
Response Variables ^a,b^	Positive Control	0.00	0.15	0.30	0.45	SEM ^c^	*p*-Value	Lin	Quad	Cub
10 days of age										
Dry matter %	32.24	31.65	31.19	31.91	31.71	0.259	0.13	0.51	0.74	0.14
Dry matter, g	1.42	1.49	1.46	1.47	1.46	0.051	0.93	0.73	0.89	0.86
Ashes, %	35.81	37.41	36.22	36.26	36.60	0.442	0.14	0.23	0.10	0.65
Ashes, g	0.51	0.56	0.53	0.53	0.54	0.021	0.67	0.50	0.45	0.71
24 days of age										
Dry matter %	38.10	38.19	38.36	38.43	38.24	0.243	0.89	0.83	0.47	0.90
Dry matter, g	3.22	3.30	3.34	3.29	3.29	0.076	0.89	0.81	0.83	0.68
Ashes, %	39.76	39.66	39.37	39.70	40.93	0.637	0.47	0.18	0.27	0.93
Ashes, g	1.28	1.31	1.32	1.31	1.35	0.036	0.82	0.53	0.62	0.66
38 days of age										
Dry matter %	42.15	42.09	42.49	41.71	41.63	0.467	0.69	0.32	0.62	0.38
Dry matter, g	7.78	7.46	7.75	7.56	7.76	0.175	0.61	0.38	0.83	0.28
Ashes, %	36.86	35.11	35.02	35.51	34.82	0.698	0.24	0.91	0.69	0.60
Ashes, g	2.86	2.63	2.70	2.68	2.70	0.070	0.18	0.59	0.72	0.69

^a^ Data are means of eight replications per treatment with pooled data of three birds in each replication. ^b^ There were no statistical differences among treatments, *p* > 0.05. ^c^ Standard error of the mean.

**Table 5 animals-11-03199-t005:** Lactic acid bacteria count, *Eimeria* oocyst excretion and antioxidant status of the breast meat of broilers added with increasing levels of humic substances in the feed, probability level, and type of effect associated with humic substances.

		Level of Humic Substances, %						Effect of Humic Substances		
Response Variables ^a^	Positive Control	0.00	0.15	0.30	0.45	SEM ^b^	*p*-Value	Lin	Quad	Cub
Lactic acid bacteria, CFUlog_10_										
10 days of age	7.40 ^c^	7.73 ^c^	7.82 ^c^	7.68 ^c^	8.41 ^d^	0.218	0.05	0.05	0.18	0.32
24 days of age	7.48	7.87	8.00	7.72	7.54	0.333	0.78	0.43	0.65	0.75
38 days of age	7.70	8.09	7.84	7.95	7.92	0.273	0.88	0.75	0.70	0.68
*Eimeria* oocyst number/g of feces										
10 days of age	62.50	156.25	181.25	43.75	118.75	64.127	0.35	0.38	0.69	0.19
24 days of age	25.00	125.00	106.25	81.25	75.00	40.153	0.83	0.77	0.75	0.89
38 days of age	368.75	431.25	562.5	345.00	731.25	168.805	0.43	0.77	0.88	0.07
Water holding capacity and antioxidant status of the breast meat										
pH	6.36	6.41	6.44	6.40	6.47	0.038	0.06	0.16	0.84	0.06
Dripping water lost, %	1.11	0.94	0.80	1.06	1.11	0.144	0.47	0.21	0.47	0.31
Centrifugation water lost, %	12.62	9.95	9.70	11.32	12.27	1.370	0.45	0.15	0.65	0.67
DPPH, mmol Trolox/kg meat	188.52 ^c,d^	213.31 ^d^	120.34 ^c^	165.61 ^c,d^	132.75 ^c^	29.833	0.05	0.05	0.19	0.10
FRAP, mmol Trolox/kg meat	2.82	2.73	2.69	2.64	2.32	0.301	0.81	0.40	0.68	0.86
TBARS, mg MDA/kg meat	0.09	0.10	0.10	0.07	0.09	0.014	0.61	0.46	0.57	0.19

^a^ Data are means of eight replications per treatment with pooled data of three birds in each replication. ^b^ Standard error of the mean. ^c,d^ Values within rows with different superscripts differ significantly, *p* < 0.05. DPPH = Scavenging activity of 1,1-diphenyl-2-picrylhydrazyl radicals; FRAP = antioxidant power of ferric radicals; TBARS = thiobarbituric acid-reactive substances.

## Data Availability

The data that support the findings of this study are available from the corresponding author upon reasonable request.

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
