# Peer review of "Addition of Different Levels of Humic Substances Extracted from Worm Compost in Broiler Feeds"

_animals, 2021, doi:10.3390/ani11113199_

Round 1

Reviewer 1 Report

animals-1425056

The manuscript submitted for review is interesting and focuses on the important issue of using alternatives to GPA in broiler chicken nutrition. However, the manuscript needs a major revision due to several uncertainties, which are listed below:

  1. What was the reason for sampling for analysis on day 38 of broiler chicken rearing? Would it not have been better to perform them on day 42, when the experiment was completed and the breast muscles were collected to determine antioxidant capacity of the meat? This makes it difficult to compare the results obtained with studies by other authors.
  2. What was the purpose of evaluation of carcass yield of chickens at 10 and 24 days of age?
  3. Part of paragraph 3 Sample Collection and Laboratory Determinations was repeated twice (Lines 129-143 and Lines 149-162)
  4. The manuscript I received lacks access to Supplementary Materials, which contain tables with important data on chicken carcass yield and tibia chracteristic. I have not had the opportunity to review them. Nevertheless, such important results should be included in the main part of the manuscript.
  5. The authors did not perform an economic analysis, therefore the results presented in the manuscript do not entitle them to draw conclusions in the present form.

Author Response

Comments to reviewers

Reviewer 1

The manuscript submitted for review is interesting and focuses on the important issue of using alternatives to GPA in broiler chicken nutrition. However, the manuscript needs a major revision due to several uncertainties, which are listed below:

  1. What was the reason for sampling for analysis on day 38 of broiler chicken rearing? Would it not have been better to perform them on day 42, when the experiment was completed and the breast muscles were collected to determine antioxidant capacity of the meat? This makes it difficult to compare the results obtained with studies by other authors.

Comments: In two previous published studies, broilers fed HS extracted from worm compost had higher carcass yield. Broilers were fed HS from 14 to 35 d of age in the first study, and 22 to 42 d of age in the second. In another recent study, carcass weight was found to be higher in HS-fed broilers aged 6-24 d. We expected to see the same effect at 38 d of age based on these findings. At 42 d of age, in addition to sampling the breast meat, crop, gizard, intestinal, and ceca content were collected for IgA and viscocity determinations; segments of the duodenum, jejeunum, and ileon were collected for histology evaluation; and samples of the intestinal mucosa were collected for microbioma analysis.

2. What was the purpose of evaluation of carcass yield of chickens at 10 and 24 days of age?

Comments: As previously stated, improved carcass yield or weight were observed in HS-fed broilers given HS at various ages and times. We wanted to see how quickly improved carcass or breast weight and yield could be seen when HS are fed from d one of age on.

3. Part of paragraph 3 Sample Collection and Laboratory Determinations was repeated twice (Lines 129-143 and Lines 149-162)

Comment: The paragraph in Lines 149-162 was removed.

4. The manuscript I received lacks access to Supplementary Materials, which contain tables with important data on chicken carcass yield and tibia chracteristic. I have not had the opportunity to review them. Nevertheless, such important results should be included in the main part of the manuscript.

Comments: Tables with carcass traits and tibia characteristics were included in the main document, and thus the Supplementary Materials were removed.

5. The authors did not perform an economic analysis, therefore the results presented in the manuscript do not entitle them to draw conclusions in the present form.

Comments: The sentence referring to economic benefits in lines 366-370 has been removed.

Reviewer 2 Report

I read this manuscript for possible publication in Animals. This paper deals with different levels of humic substances as growth promoters in broiler feeds. The authors of this paper have taken a very important and topical issue which is the use of growth promoters in feeding broilers. Given the seriousness of this problem, this paper provides valuable information  in this field. I consider it appropriate to taking this type of study.
It should be noted that the manuscript is written generally correctly and clearly and falls in the scope of Animals. The strategy and methodology of the research is correct. The obtained results are properly described and interpreted against the background of world literature. 

Reading this text I would only suggest:
- shorten title with the words: "extracted from worm compost"
- removing the repeating paragraph from 148 to 163 lines
- perhaps shorten too detail described material and methods
- adding the table number, which is missing on line 249
- changing the conclusions, because they are more focused on the costs of 
  using HS and not enough on the positive effects of this feed additive

This paper should be considered as acceptable for publication in Animals, because this paper is a significant and important contribution to the field of investigation.

Author Response

Comments to reviewers

Reviewer 2

Reading this text I would only suggest:
- shorten title with the words: "extracted from worm compost" :

Comment: the title was shortened as follows “Addition of different levels of humic substances extracted from worm compost in broiler feeds”

- removing the repeating paragraph from 148 to 163 lines

Comment: the paragraph was removed

- perhaps shorten too detail described material and methods

Comment: the Editorial recommendation was to check for overlaping in the Matherial and methods section

- adding the table number, which is missing on line 249

Comment: the Table number was written.

- changing the conclusions, because they are more focused on the costs of 
  using HS and not enough on the positive effects of this feed additive
Comment: The sentence referring to economic benefits in lines 366-370 has been removed.

Reviewer 3 Report

The present paper has several flaws, however the manuscript may be published after completing the deficiencies and refining the details.

All sections  need to be corrected.

I suggest to improve the following formulations in the text.

In the present paper, authors did not follow the “Instructions for Authors” e.g.: Abbreviations should be defined the first time they appear in each of three sections: the abstract; the main text; the first figure or table. For example line 32, 35, … Please revise this.

Introduction:

Line 41-56: Paragraph is too long. Please shorten it.

Line 65-72: You should describe the effects of specific HS components on the animals based on the literature. Please provide more information on the use of HS in animal testing.

Material and Methods

Provide the approval number of your local animal committee for all procedures used in your research, or demonstrate compliance with the standard procedures of the universities / institutes where the research was conducted.

In section 2.2,  you should describe the stocking density per 1m2 on the 42nd day of experiment.

Line 79: How did you  prepare worm compost? Which species of worms did you use? How long did you compost it? Please revise this.

In section 2.3, the first paragraph is replication (see line 128-143 and line 148-164). Please change it.

Results and Tables

Why didn’t you provide “n=…”? Please describe all the tables correctly.

Discussion

Line 318: Change „Trypanosoma brucei brucei brucei” to „Trypanosoma brucei brucei

Conclusions

Line 370-371: In your study, LAB was statistically significantly higher only in group 0.45% HS, but in conclusions you wrote:  “Another positive effect of HS was the increased 
LAB counts in 10-d-old chicks”. Please correct it and specify it.

Author Response

Comments to reviewers

Reviewer 3

In the present paper, authors did not follow the “Instructions for Authors” e.g.: Abbreviations should be defined the first time they appear in each of three sections: the abstract; the main text; the first figure or table. For example line 32, 35, … Please revise this.

Comment: All abbreviations were defined.

Introduction:

Line 41-56: Paragraph is too long. Please shorten it.

Comment: the paragraph was splitted

Line 65-72: You should describe the effects of specific HS components on the animals based on the literature. Please provide more information on the use of HS in animal testing.

Comment: A more detailed paragraph about the effects of blends of HS was added.

Material and Methods

Provide the approval number of your local animal committee for all procedures used in your research, or demonstrate compliance with the standard procedures of the universities / institutes where the research was conducted.

Comment: The approval folio number was provided in Materials and Methods.

In section 2.2,  you should describe the stocking density per 1mon the 42nd day of experiment.

Comment: the stocking density was provided.

Line 79: How did you  prepare worm compost? Which species of worms did you use? How long did you compost it? Please revise this.

Comment: a description of the worm compost production was included

In section 2.3, the first paragraph is replication (see line 128-143 and line 148-164). Please change it.

Comment: the paragraph was deleted.

Results and Tables

Why didn’t you provide “n=…”? Please describe all the tables correctly.

Comment: The number of replications per treatment was provided in the footnotes of all result tables.

Discussion

Line 318: Change „Trypanosoma brucei brucei brucei” to „Trypanosoma brucei brucei

Comments: the change was made.

Conclusions

Line 370-371: In your study, LAB was statistically significantly higher only in group 0.45% HS, but in conclusions you wrote:  “Another positive effect of HS was the increased 
LAB counts in 10-d-old chicks”. Please correct it and specify it.

Comments: the statement was corrected

Reviewer 4 Report

General comments

The paper deals with the  effect of Humic substances extracted from worm compost on productive performance of broiler chickens.  This is a very interesting topic and the paper can provide some new information about the use of these substances as growth promoters. The work is within the scope of the journal and presented some novelty. The manuscript in general is sound and is well written. The experimental design is correct. I have only some doubts and minor suggestions about the manuscript

Specific comments

Introduction

In the introduction section it should be justified more clearly what is the new information the paper provides compared to previous studies, in particular compared to the reference number 18.

Material and Methods

Authors should include the ethical consideration statement for the use of animals for experimental pourposes and the number of the approval protocol.

L120: Diets were fed in mash form during the whole  productive cycle? Not in pelleted form at the end?. This could have and effect on the low feed intake recorded in all groups and the low weigh gain recorded at day 42 in all grooups, compared to the standars for Ross308?. Authors could include this point in the discussion section, may the results should be different otherwise?

L120: Please explain how HS were added to the diets- top-dressed?, daily?....

L148-162: this paragraph is repeated

L137: why samples of jejunum contents and not cecum contents were taken for LAB determinations?. Please indicate how samples were kept (-20, -80ºC) until analisys.

L176: indicate the Brand, country etc for the count chamber

L192: include here the reference ([20])

L193: include here the reference ([22]; revcise the order of the references, since reference [21] appears before thge reference [20]

L203: was this spectrophotometer used for all the analyses: DPPH, FRAP, TBARS?

L209: -20ºC and not ~20ºC

L214: include the package or statistical programme used to perform the analysis (reference); Describe the used statistical model, what are the fixed effects and random if exist.

Results

Taking into account the design of the study I recomend to describe the observed differences between the positive control and the rest of treatments and then comment on the polynomial contrasts (0:negative control; 15, 30 and 45 of HS). Because otherwise sometimes is confusing, because individual comparisons and trends presented similar information and sounds redundant. In my opinion polynomial contrast shows better the information and should be the preferred statistic. However I understand that as authors have included a positive control some other comparison could be analysed.

L225: describe what occured in the second period (d15-28)

Tables 2 and 3. Used P-value instead of p<

Table 3. revise superscripts for DPPH and the description of differences in L250; delete (e) from the footnote

Discussion

L318: Trypanosoma brucei brucei instead of Trypanosoma brucei brucei brucei

I suggest to include some discussion about the mortality results observed with the HS

Conclusion

L366-369: this results did not appeared in the manuscript, neither the calcuation of the production costs….

Author Response

Comments to reviewers

Reviewer 4

Specific comments

Introduction

In the introduction section it should be justified more clearly what is the new information the paper provides compared to previous studies, in particular compared to the reference number 18.

Comment: The justification for the study was added

Material and Methods

Authors should include the ethical consideration statement for the use of animals for experimental pourposes and the number of the approval protocol.

Comment: The approval folio number was provided in Materials and Methods

L120: Diets were fed in mash form during the whole  productive cycle? Not in pelleted form at the end?. This could have and effect on the low feed intake recorded in all groups and the low weigh gain recorded at day 42 in all grooups, compared to the standars for Ross308?. Authors could include this point in the discussion section, may the results should be different otherwise?

Comment: mash feed was available during the entire production trial.

L120: Please explain how HS were added to the diets- top-dressed?, daily?....

Comment: HS were added at the top of the diets.

L148-162: this paragraph is repeated

Comment: the paragraph was deleted.

L137: why samples of jejunum contents and not cecum contents were taken for LAB determinations?. Please indicate how samples were kept (-20, -80ºC) until analisys.

Comment: The Firmicutes phyla dominate the bacterial communities in the small intestine, with the Lactobacillus genus accounting for 70% of the total, while the jejunum is responsible for the majority of nutrient digestion and absorption. The aim of measuring LAB in the jejunum was to link increases in productivity to an increase in LAB counts, which is a beneficial microbe for intestinal health.

Following the collection of samples, they were immediately transported to the lab for culturing.

L176: indicate the Brand, country etc for the count chamber

Comment: the information has been updated

L192: include here the reference ([20])

Comment: the reference was re-located as requested

L193: include here the reference ([22]; revcise the order of the references, since reference [21] appears before thge reference [20]

Comments: the reference was re-located as requested

Reference [20] first appears on L183 and then L197; while Reference [21] shows in L186

L203: was this spectrophotometer used for all the analyses: DPPH, FRAP, TBARS?

Comment: The same spectrophotometer was used for all analyses

L209: -20ºC and not ~20ºC

Comment: the correction was made

L214: include the package or statistical programme used to perform the analysis (reference); Describe the used statistical model, what are the fixed effects and random if exist.

Comment: the information was added.

Results

Taking into account the design of the study I recomend to describe the observed differences between the positive control and the rest of treatments and then comment on the polynomial contrasts (0:negative control; 15, 30 and 45 of HS). Because otherwise sometimes is confusing, because individual comparisons and trends presented similar information and sounds redundant. In my opinion polynomial contrast shows better the information and should be the preferred statistic. However I understand that as authors have included a positive control some other comparison could be analysed.

Comments: The recommendation was taken into account and the description of results was modified.

Plynomial contrasts were actually used to compare the type of response associated with the addition of HS. The appropriate correction was made.

L225: describe what occured in the second period (d15-28)

Comment: In the second paragraph of the Discussion this topic was addressed

Tables 2 and 3. Used P-value instead of p<

Comment: The correction was made

Table 3. revise superscripts for DPPH and the description of differences in L250; delete (e) from the footnote

Comment: the superscripts for DPPH at 0.45% HS was corrected; e superscript was deleted.

Discussion

L318: Trypanosoma brucei brucei instead of Trypanosoma brucei brucei brucei

Comment: the change was made.

I suggest to include some discussion about the mortality results observed with the HS

Comment: In the fourth paragraph of the Discussion this topic was addressed

Conclusion

L366-369: this results did not appeared in the manuscript, neither the calcuation of the production costs….

Comment: The sentence referring to economic benefits in lines 366-370 has been removed.

Round 2

Reviewer 1 Report

Considering that the Authors have clarified all doubts and corrected the manuscript according to the instructions, I recommend the manuscript for publication in Animals.